# Proving Theorems using Incremental Learning and Hindsight Experience Replay

## Abstract

Traditional automated theorem provers for first-order logic depend on speed-optimized search and many handcrafted heuristics that are designed to work best over a wide range of domains. Machine learning approaches in literature either depend on these traditional provers to bootstrap themselves or fall short on reaching comparable performance. In this paper, we propose a general incremental learning algorithm for training domain-specific provers for first-order logic without equality, based only on a basic given-clause algorithm, but using a learned clause-scoring function. Clauses are represented as graphs and presented to transformer networks with spectral features. To address the sparsity and the initial lack of training data as well as the lack of a natural curriculum, we adapt hindsight experience replay to theorem proving, so as to be able to learn even when no proof can be found. We show that provers trained this way can match and sometimes surpass state-of-the-art traditional provers on the TPTP dataset in terms of both quantity and quality of the proofs.

## 1 Introduction

> *I believe that to achieve human-level performance on hard problems, theorem provers likewise must be equipped with soft knowledge, in particular soft knowledge automatically gained from previous proof experiences. I also suspect that this will be one of the most fruitful areas of research in automated theorem proving. And one of the hardest.* Schulz (2017, E's author)

Automated theorem proving (ATP) is an important tool both for assisting mathematicians in proving complex theorems as well as for areas such as integrated circuit design, and software and hardware verification (Leroy, 2009; Klein, 2009). Initial research in ATP dates back to 1960s (*e.g.*, Robinson (1965); Knuth & Bendix (1970)) and was motivated partly by the fact that mathematics is a hallmark of human intelligence. However, despite significant research effort and progress, ATP systems are still far from human capabilities (Loos et al., 2017). The highest performing ATP systems (*e.g.*, Cruanes et al. (2019); Kovács & Voronkov (2013)) are decades old and have grown to use an increasing number of manually designed heuristics, mixed with some machine learning, to obtain a large number of search strategies that are tried sequentially or in parallel. Recent advances (Loos et al., 2017; Chvalovskỳ et al., 2019) build on top of these provers and used modern machine learning techniques to augment, select or prioritize their heuristics, with some success. However, these machine-learning based provers usually require initial training data in the form of proofs, or positive and negative examples (provided by the high-performing existing provers) from which to bootstrap. Recent works do not build on top of other provers, but still require existing proof examples (*e.g.*, Goertzel (2020); Polu & Sutskever (2020)).

Our perspective is that the reliance of current machine learning techniques on high-end provers limits their potential to consistently surpass human capabilities in this domain. Therefore, in this paper we start with only a basic theorem proving algorithm, and develop machine learning methodology for bootstrapping automatically from this prover. In particular, given a set of conjectures without proofs, our system trains itself, based on its own attempts and (dis)proves an increasing number of conjectures, an approach which can be

viewed as a form of incremental learning. A particularly interesting recent advance is rlCop (Kaliszyk et al., 2018; Zombori et al., 2020), which is based on the minimalistic leanCop theorem prover (Otten & Bibel, 2003), and is similar in spirit to our approach. It manages to surpass leanCop's performance—but falls short of better competitors such as E, likely because it is based on a 'tableau' proving style rather than a saturation-based one. This motivated Crouse et al. (2021) to build on top of a saturation-based theorem prover and indeed see some improvement, while still not quite getting close to E.

However, all previous approaches using incremental learning have a blind spot, because they learn exclusively from the proofs of successful attempts: If the given set of conjectures have large gaps in their difficulties, the system may get stuck at a suboptimal level due to the lack of new training data. This could in principle even happen at the very start, if all theorems are too hard to bootstrap from. To tackle this issue, Aygün et al. (2020) propose to create synthetic theorem generators based on the axioms of the actual conjectures, so as to provide a large initial training set with diverse difficulty. Unfortunately, synthetic theorems can be very different from the target conjectures, making transfer difficult.

In this paper, we adapt instead hindsight experience replay (HER) (Andrychowicz et al., 2017) to ATP: clauses reached in proof attempts are turned into goals in hindsight. This generates a large amount of "auxiliary" theorems with proofs for the learner, even when no theorem from the original set can be proven.

We compare our approach on a subset of TPTP (Sutcliffe, 2017) with the state-of-the-art E prover (Schulz, 2002; Cruanes et al., 2019), which performs very well on this dataset. Our learning prover eventually reaches equal or better performance on 16 domains out of 20. In addition, it finds shorter proofs than E in approximately 98% of the cases. We perform an ablation experiment to highlight specifically the role of hindsight experience replay. In the next sections, we explain our incremental learning methodology with hindsight experience replay, followed by a description of the network architecture and experimental results.

## 2 Methodology

For the reader unfamiliar with first-order logic, we give a succinct primer in Appendix A. From an abstract viewpoint, in our setting the main object under consideration is the *clause*, and two operations, *factoring* and *resolution*. These operations produce more clauses from one or two parent clauses. Starting from a set of axiom clauses and negated conjecture clauses (which we will call *input clauses*), the two operations can be composed sequentially to try to reach the *empty clause*, in which case its ancestors form a refutation proof of the input clauses and correspond to a proof of the non-negated conjecture. We call the `tree_size` of a clause the number of symbols (with repetition) appearing in the clause; for example `tree_size`$(p(X, a, X, b) \lor q(a))$ is 7.

We start by describing the search algorithm, which allows us then to explain how we integrate machine learning and to describe our overall incremental learning system.

### 2.1 Search algorithm

To assess the incremental learning capabilities of recent machine learning advances, we have opted for a simple base search algorithm (see also Kaliszyk et al. (2018) for example), instead of jump-starting from an existing high-performance theorem prover. Indeed, E is a state-of-art prover that incorporates a fair number of heuristics and optimizations (Schulz, 2002; Cruanes et al., 2019), such as: axiom selection, simplifying the axioms and input clauses, more than 60 literal selection strategies, unit clause rewriting, multiple fast indexing techniques, clause evaluation heuristics (tree size preference, age preference, preference toward symbols present in the conjecture, watch lists, etc.), strategy selection based on the analysis of the prover on similar problems, multiple strategy scheduling with 450 strategies tuned on TPTP 7.3.0,[1] integration of a PicoSAT solver, etc. Other machine learning provers

---

[1]See `http://www.tptp.org/CASC/J10/SystemDescriptions.html#E---2.5`.

based on E (*e.g.*, Jakubův et al. (2020); Loos et al. (2017)) automatically take advantage of at least some of these improvements (but see also Goertzel (2020)).

Like E and many other provers, we use a variant of the DISCOUNT loop (Denzinger et al., 1997), itself a variant of the given-clause algorithm (McCune & Wos, 1997) (See Algorithm 3 in Appendix B). The input clauses are initially part of the `candidates`, while the `active_clauses` list starts empty. A candidate clause is selected at each iteration. New factors of the clause, as well as all its resolvents with the active clauses are pushed back into `candidates`. The given clause is then added to `active_clauses`, and we say that it has been *processed*. We also use the standard forward and backward subsumptions, as well as tautology deletion, which allow to remove too-specific clauses that are provably unnecessary for refutation. If `candidates` becomes empty before the empty clause can be produced, the algorithm returns `"saturated"`, which means that the input clauses are actually counter-satisfiable (the original conjecture is dis-proven).

Given-clause-based theorem provers often use several priority queues to sort the set of candidates (*e.g.*, McCune & Wos (1997); Schulz (2002); Kovács & Voronkov (2013)). We also use three priority queues: the age queue, ordered from the oldest generated clause to the youngest, the weight queue, ordered by increasing `tree_size` of the clauses, and the learned-cost queue, which uses a neural network to assign a score to each generated clause. The age queue ensures that every generated clause is processed after a number of steps that is at most a constant factor times its age. The weight queue ensures that small clauses are processed early, as they are "closer" to the empty clause. The learned-cost queue allows us to integrate machine learning into the search algorithm, as detailed below.

## 2.2 Clause-scoring network and hindsight experience replay

The clause-scoring network can be trained in many ways so as to find proofs faster. A method utilized by Loos et al. (2017) and Jakubuv & Urban (2019) turns the scoring task into a classification task: a network is trained to predict whether the clause to be scored will appear in the proof or not. In other words, the probability predicted by an 'in-proofness' classifier is used as the score. To train, once a proof is found, the clauses that participate in the proof (*i.e.*, the ancestors of the empty clause) are considered to be positive examples, while all other generated clauses are taken as negative examples.[2] Then, given as input one such generated clause $x$ along with the input clauses $C_s$, the network must learn to predict whether $x$ is part of the (found) proof.

There are two main drawbacks to this approach. First, if all conjectures are too hard for the initially unoptimized prover, no proof is found and no positive examples are available, making supervised learning impossible. Second, since proofs are often small (often a few dozen steps), only few positive examples are generated. As the number of available conjectures is often small too, there is far too little data to train a modern high-capacity neural network. Moreover, for supervised learning to be successful, the conjectures that can be proven must be sufficiently diverse, so the learner can steadily improve. Unfortunately, there is no guarantee that such a curriculum is available. If the difficulty suddenly jumps, the learner may be unable to improve further. These shortcomings arise because the learner only uses successful proofs, and all the unsuccessful proof attempts are discarded. In particular, the overwhelming majority of the generated clauses become negative examples, and most need to be discarded to maintain a good balance with the positive examples.

To leverage the data generated in unsuccessful proof attempts, we adapt the concept of hindsight experience replay (HER) (Andrychowicz et al., 2017) from goal-conditioned reinforcement learning to theorem proving. The core idea of HER is to take any "unsuccessful" trajectory in a goal-based task and convert it into a successful one by treating the final state as if it were the goal state, in hindsight. A deep network is then trained with this trajectory, by contextualizing the policy with this state instead of the original goal. This way, even in

---

[2]These examples are technically not necessarily negative, as they may be part of another proof. But avoiding these examples during the search still helps the system to attribute more significance to the positive examples.

the absence of positive feedback, the network is still able to adapt to the *domain*, if not to the goal, thus having a better chance to reach the goal on future tries.

Inspired by HER, we use the clauses generated during *any* proof attempt as additional conjectures, which we call *hindsight goals*, leading to a supply of positive and negative examples. Let $D$ be any non-input clause generated during the refutation attempt of $C_s$. We call $D$ a *hindsight goal*.[3] Then, the set $C_s \cup \{\neg D\}$ can be refuted. Furthermore, once the prover reaches $D$ starting from $C_s \cup \{\neg D\}$, only a few more resolution steps are necessary to reach the empty clause; that is, there exists a refutation proof of $C_s \cup \{\neg D\}$ where $D$ is an ancestor of the empty clause. Hence, we can use the ancestors of $D$ as positive examples for the negated conjecture and axioms $C_s \cup \{\neg D\}$. This generates a very large number of examples, allowing us to effectively train the neural network, even with only a few conjectures at hand.

Since each domain has its own set of axioms, and a separate network is trained per domain, axioms are not provided as input to the scoring network. Although the set of active clauses is an important factor in determining the usefulness of a clause, we ignore it in the network input to keep the network size smaller.

### 2.3 Incremental learning algorithm

Typical supervised learning ATP systems require a set of proofs (provided by other provers) to optimize their model (*e.g.*, Loos et al. (2017); Jakubův et al. (2020); Aygün et al. (2020)). Success is assessed by cross-validation. In contrast, we formulate ATP as an incremental learning problem—see in particular Orseau & Lelis (2021); Jabbari Arfaee et al. (2011). Given a pool of unproven conjectures, the objective is to prove as many as possible, even using multiple attempts, and ideally as quickly as possible. Hence, the learning system must bootstrap directly from initially-unproven conjectures, without any initial supervised training data. Success is assessed by the number of proven conjectures, and the time spent solving them. Hence, we do not need to split the set of conjectures into train/test/validate sets because, if the system overfits to the proofs of a subset of conjectures, it will not be able to prove more conjectures.

Our incremental learning system is described in Algorithm 1. Initially, all conjectures are unproven and the clause-scoring network is initialized randomly. At this stage, we have no information on how long it takes to prove a certain conjecture, or whether it can be proven at all. The prover attempts to prove all conjectures provided using a scheduler (described below), so as to vary time limits for each conjecture. This ensures that proofs for easy conjectures are obtained early, and the resulting positive and negative examples are then used to train the clause-scoring network. As the network learns, more conjectures can be proven, providing in turn more data, and so on. This incremental learning algorithm thus allows us to automatically build a capable prover for a given domain, starting from a basic prover that may not even be able to prove a single conjecture in the given set.

**Time scheduling.** All conjectures are attempted in parallel, each on a CPU. For each conjecture, we use the uniform budgeted scheduler (UBS) algorithm (Helmert et al., 2019, section 7) to further simulate running in (pseudo-)parallel the solver with varying time budgets, and restarting each time the budget is exhausted. In the terminology of UBS, we take $T(k, r) = 3r2^{k-1}$ in seconds, but we cap $k \leq k_{\max} = 10$. A UBS instance simulates on a single CPU running $k_{\max}$ restarting programs, by interleaving them: On a 'virtual' CPU of index $k \in \{1, \ldots, k_{\max}\}$, a program corresponds to running the prover for a budget of $3 \cdot 2^{k-1}$ seconds before restarting it for the same budget of time and so on; $r$ is the number of restarts. Hence, as the network learns, each conjecture is incrementally attempted with time budgets of varying sizes (3s, 6s, 12s, . . . , 3072s), using no more than one hour, while carefully balancing the cumulative time spent within each budget (Luby et al., 1993; Helmert et al., 2019). Once a proof has been found for a conjecture, the scheduler is not stopped, so as to continue searching for more (often shorter) proofs.

---

[3]Note that, while the original version of HER (Andrychowicz et al., 2017) only uses the last reached state as a single hindsight goal, we use all intermediate clauses, providing many more data points.

**Algorithm 1** Distributed incremental learning. `launch` starts a new process in parallel. For each conjecture an instance of UBS decides the sequence of time limits for solving attempts.

```
def main(conjectures):
  # Launch and connect learners, actors and manager with example buffer &  task queue
  example_buffer = create_example_buffer()
  task_queue = create_task_queue()
  learners = [for i = 1..10: launch learner(example_buffer)]
  for i = 1..1000: launch actor(task_queue, learners, example_buffer)
  actor_manager = launch actor_manager(conjectures, task_queue)
  wait for actor_manager to finish

def learner(example_buffer):
  repeat forever:
    # Sample a batch of examples and train the network.
    batch = sample_batch_uniformly(example_buffer)
    minimize_classification_loss(batch)  # we use cross-entropy

def actor(task_queue, learners, example_buffer)
  repeat forever:
    # Fetch a task and attempt to prove the conjecture.
    conjecture, time_limit = get_task(task_queue)
    learner = sample_uniformly(learners)
    call search(conjecture) for at most time_limit seconds # see Alg. 3 Appendix B
      and obtain generated_clauses
    examples = sample_examples(generated_clauses) # see Alg. 2
    put_examples(example_buffer, examples)

def actor_manager(conjectures, task_queue):
  schedulers = []
  for conjecture in conjectures:
    schedulers[conjecture] = initialize_UBS() # see Section 2.3
  repeat until all conjectures have been proven:
    # Choose a random conjecture and enqueue it.
    conjecture = sample_uniformly(conjectures)
    scheduler = schedulers[conjecture]
    time_limit = get_next_time_limit(scheduler)
    put_task(task_queue, (conjecture, time_limit))
```

**Distributed implementation.** Our implementation consists of multiple actors running in parallel, a manager that distributes tasks to the actors using the time scheduling algorithm, and a task queue that handles manager-actors communication. We used ten learners training ten separate models to increase the diversity of the search without having to increase the number of actors. These learners are fed with training examples from the actors and use them to update their parameters of their clause-scoring networks. Note that during the first 1 000 updates, the actors do not use the clause-scoring network as its outputs are mostly random.[4]

**Subsampling hindsight goals and examples.** With HER, the number of available examples is actually far too large: if, after a proof attempt, $n$ clauses have been generated ($n$ may be in the thousands), not only can each clause be used as a hindsight goal, but there are about $n^2$ pairs (positive example, hindsight goal), and far more negative examples. This suddenly puts us in a very data-rich regime, which contrasts with the data scarcity of learning only from proofs of the given conjecture. Hence, we need to *subsample* the examples to prevent overwhelming the learner (see Algorithm 2 in the appendix). To this end, we first estimate the number of examples the learner can consume per second before sampling. But there is an additional difficulty: the number of possible clauses is exponentially large in the `tree_size` of the clause, while small clauses are likely more relevant since the empty clause

---

[4]We picked 1000 as it appeared to be approximately the number of steps required for the learner to reach the base prover performance on a few experiments.

---

**Algorithm 2** Example sampling algorithm.

---

```
def sample_examples(generated_clauses):
  # Estimate the number of examples that can be consumed by the learner
  target_num_examples =
    time_elapsed_since_last_attempt × target_num_examples_per_second

  # Remove the input clauses
  hindsight_goals = generated_clauses \ input_clauses

  # Subsample the goals and the examples
  examples = []
  sizes = {tree_size(c) : c ∈ hindsight_goals}
  for size in sizes:
    size_goals = {c ∈ hindsight_goals : tree_size(c) == size}
    w_size = 1 / ln(size + e) - 1 / ln(size + e + 1) # See Appendix C
    num_examples = ceil(target_num_examples × w_size)
    for _ in range(num_examples):
      goal = uniform_sample(size_goals) # pick hindsight goal of this size
      anc = ancestors(goal)
      examples += [positive_example(uniform_sample(anc), goal)]
      examples += [negative_example(uniform_sample(hindsight_goals \ anc), goal)]
  return examples
```

---

(which is the true target) has size 0. Moreover, clauses can be rather large: a `tree_size` over 300 is quite common, and we observed some `tree_size` over 6 000. To correct for this, we fix the proportion of positive and negative examples for each hindsight goal clause size, ensuring that small hindsight goal clauses are favoured, while allowing a diverse sample of large clauses, using a heavy-tail distribution $w_s$ described in Appendix C. Finally, all the positive and negative examples thus sampled are added to the training pool for the learners.

## 2.4 REPRESENTATION

Our clause scoring network receives as input the clause to score, $x$, the hindsight goal clause, $g$, and a sequence of negated conjecture clauses $C_s$. Individual clauses are transformed to directed acyclic graphs (an example is depicted in Figure 1) with five different node types. First, there is a clause node, whose children are literal nodes, corresponding to all literals of the clause (each one is associated with a predicate). The children of literal nodes represent the arguments of the predicate; they are either variable-term nodes if the argument is a variable, or atomic-term nodes otherwise[5]. Children of atomic-term nodes follow the same description. Finally, each variable-term node is linked to a variable node, which has as many parents as there are instances of the corresponding variable in the clause.

To each node, we associate a feature vector composed of the following five components: (i) A one-hot vector of length 3, encoding if the node belongs to $x$, $g$ or a member of $C_s$. (ii) A one-hot vector of length 5 encoding the node type: clause, literal, atomic-term, variable-term or variable. (iii) A one-hot vector of length 2 encoding if the node belongs to a positive or negative literal (null vector for clause and variable nodes). (iv) A hash vector representing the predicate name or the function/constant name respectively for predicate or atomic-term nodes (null vector for other nodes). (v) A hash vector representing the predicate/function argument slot in which the term is present (null vector for clause, literal and variable nodes). Hash vectors are randomly sampled uniformly on the 64 dimensional unit hyper-sphere, using the name of the predicate, function or constant (and the argument position for slots) as seed.

The node feature vectors are projected into a 64-dimensional node embedding space using a linear layer that trains during learning. We use a Transformer encoder architecture (Vaswani et al., 2017) for the clause-scoring network, whose input is composed of the set of node

---

[5] A constant argument is equivalent with a function of arity 0.

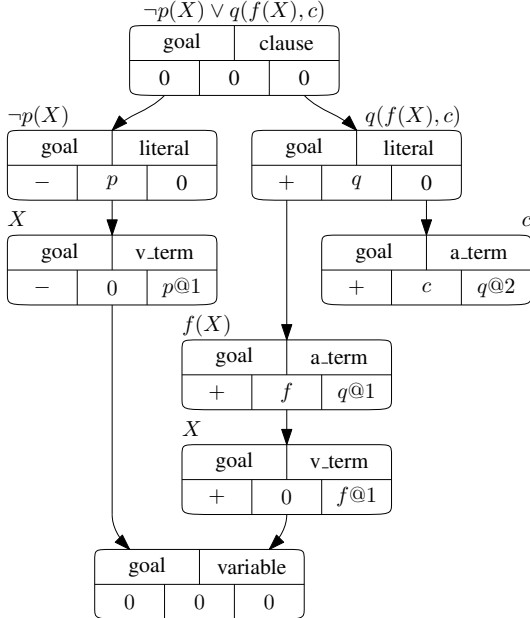

Figure 1: Clause graph of a goal clause. Each node has five features: clause type, node type, literal polarity, symbol hash and argument slot hash. The parts of formula corresponding to each node are shown outside of the nodes.

embeddings in the current clause $x$, goal clause $g$ and conjecture clauses $C_s$, up to 128 nodes. For each node, we compute a spectral encoding vector representing its position in the clause graph (Dwivedi & Bresson, 2020); this is given by the eigenvectors of the Laplacian matrix of the graph from which we keep only the 64 first dimensions, corresponding to the low frequency components. It replaces the traditional positional encoding of Transformers. Note that if there are more than 128 nodes in the set of clause graphs, we prioritize $x$, then $g$ and $C_s$. Within each graph, we order the nodes from top to bottom then left to right (e.g. the first nodes to be filtered out would be variable- or atomic-term nodes of the last conjecture clause). We only keep the transformer encoder output corresponding to the root node of the target clause and project it, using a linear layer, into a single logit, representing the probability that $x$ will be used to reach $g$ starting from $C_s$.

## 3 Experiments

To evaluate our approach, we use the Thousands of Problems for Theorem Proving (TPTP) library (Sutcliffe, 2017) version 7.5.0. We focus on FOF and CNF domains without the equality predicate (those which use the symbol =) that refer to axioms files and contain less than 1 000 axioms. This results in 20 domains, named after the corresponding axioms files. Note that the GEO6 and GEO8 axiom files often appear together, and we group them into the GEO6 domain. The resulting list of domains and conjectures can be found in the supplementary material.

We ran our incremental learning algorithm with hindsight experience replay (IL w/HER) for seven days on the twenty domains. For each domain, we trained ten models using 1000 actors. We logged every successful attempt that lead to a proof during training, along with the time elapsed, the number of clauses generated, the length of the proof, and the proof itself.

In order to show the importance of HER in achieving the results above, we ran the same experiments without HER (IL w/o HER), training the clause-scoring network using solely the data extracted from proofs found for the input problems.

| Domain | Conjectures | Basic | IL w/o HER | IL w/HER | E (1h) | E (7d) |
|--------|-------------|-------|------------|----------|--------|--------|
| FLD1 | 135 | 9 | 12 | 66 | 69 | **75** |
| FLD2 | 143 | 6 | 8 | 86 | 87 | **97** |
| GEO6 | 164 | 86 | 164 | 164 | 164 | 164 |
| GEO7 | 38 | 36 | 38 | 38 | 38 | 38 |
| GEO9 | 37 | 37 | 37 | 37 | 37 | 37 |
| GRP5 | 10 | 6 | 10 | 10 | 10 | 10 |
| KRS1 | 41 | 9 | 37 | **41** | 40 | 40 |
| LCL3 | 65 | 33 | 60 | 60 | 60 | 60 |
| LCL4 | 168 | 35 | 120 | **155** | 134 | 153 |
| MED1 | 9 | 0 | 0 | 9 | 9 | 9 |
| NUM1 | 10 | 9 | 9 | 9 | 9 | 9 |
| NUM9 | 36 | 4 | 9 | 19 | **25** | **25** |
| PLA1 | 26 | 3 | 26 | 26 | 26 | 26 |
| PUZ4 | 7 | 1 | 2 | **5** | 4 | 4 |
| SET1 | 11 | 6 | 11 | 11 | 11 | 11 |
| SWB2 | 6 | 3 | 3 | **6** | 4 | **6** |
| SWB3 | 3 | 3 | 3 | 3 | 3 | 3 |
| SYN1 | 199 | 199 | 199 | 199 | 199 | 199 |
| SYN2 | 7 | 4 | 4 | 4 | **5** | **5** |
| TOP1 | 1 | 1 | 1 | 1 | 1 | 1 |
| Total | 1116 | 490 | 753 | 949 | 935 | **972** |

Table 1: Number of conjectures proven on the twenty domains. E results at the one hour mark (1h) are shown in addition to the E results at the full seven day mark (7d).

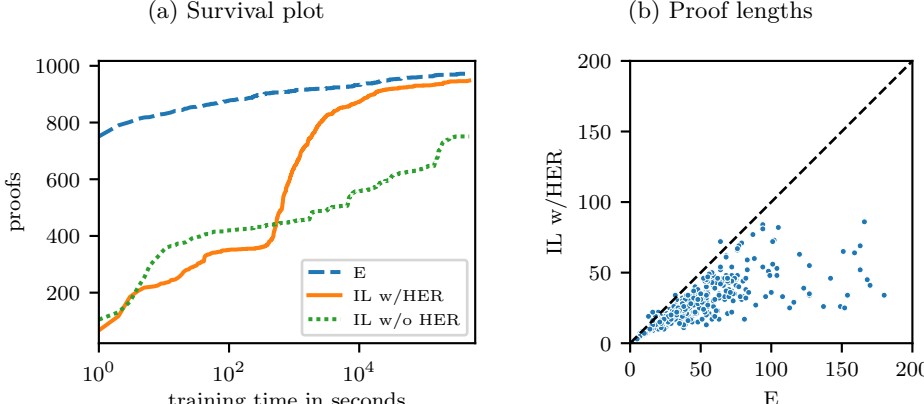

(a) Survival plot      (b) Proof lengths

Figure 2: (a) Survival plot showing the progress of E and incremental learning with and without hindsight experience replay over the course of seven days of training. The time scale is logarithmic. (b) Scatter plot of the shortest proof lengths achieved by E vs. incremental learning with hindsight experience replay on the conjectures that can be proven by both.

To compare our results to the state-of-the-art, we also ran the E prover (Cruanes et al., 2019) version 2.5 on each of the conjectures in the same domains, in "auto" and "auto-schedule" modes and with time limits of 2 days, 4 days and 7 days. We then took the union of all solved conjectures from these runs, so as to give E the best shot possible, and because E's behaviour is sensitive to the given time limit. As we never run our prover longer than one hour at a time, we measured E's performance at the one hour mark in addition to the full seven day mark.

The hyperparameters used in all experiments are given in Appendix D.

**Conjectures proven.** Table 1 shows the number of conjectures proven by our basic prover (without the learned-cost queue), IL w/o HER, IL w/HER, E at one hour mark (E 1h), and

E at full seven day mark (E 7d), as well as the actual number of conjectures, in each domain and in total. According to these results, IL w/HER proved 1.94 times as many problems as the basic prover, improving its performance by almost a factor of two. It proved 14 (1.5%) more conjectures than E 1h and 23 (2.42%) fewer conjectures than E 7d. It outperformed E 7d on three of the domains and matching its performance on 13 of the domains. Nine of the proofs found by IL w/HER were missed by E, which implies that the combination of these provers are better than either of them.

**Without hindsight.**   IL w/o HER performed significantly worse, failing to prove 198 (20.9%) of the conjectures that can be proven by IL w/HER. As expected, most of the failures happened in the "hard" domains, where the basic prover was not performing well. Without enough proofs from which to learn, IL w/o HER stalled and showed little to no progress on these domains.

**Training vs. searching.**   In Figure 2a, a comparison of the progress of E and the improvement of our systems can be seen as a survival plot over seven days of run time (wall-clock). Unlike E, which performed up to seven days of proof search per conjecture but has been under constant development for almost two decades, IL w/HER spent the same seven days to train provers based on a simple proof search algorithm from scratch, and ended up finding almost as many proofs as E.

**Quality of proofs.**   We also looked at the individual proofs discovered by both systems. Incremental learning combined with the revisiting of previously proven conjectures allowed our system to discover shorter proofs continually. On average, proofs got 13% shorter after their initial discovery. We also observed that the shortest proofs found by our system were shorter than those found by E. Out of the 941 conjectures proven by both systems, our proofs were shorter for 921 conjectures (97.9%) whereas E's proofs were shorter for only 9 conjectures (0.956%) with 11 proofs being of the same length. Figure 2b shows a scatter plot of the lengths of the shortest proofs found by E vs. found by IL w/HER.

**Speed of search.**   E was able to perform the proof search 25.6 times faster than our provers in terms of clauses generated per second. We believe that the only way for our system to compete with E under these conditions is to find scoring functions that are as strong as the numerous heuristics built into E for these domains.

## 4    CONCLUSION

In this work, we provide a method for training domain-specific theorem provers given a set of conjectures without proofs. Our proposed method starts from a very simple given-clause algorithm and uses hindsight experience replay (HER) to improve, prove an increasing number of conjectures in incremental fashion. We train a transformer network using spectral features in order to provide a useful scoring function for the prover. Our comparison with the state-of-the-art heuristic-based prover E demonstrates that our approach achieves comparable performance to E, while our proofs are almost always shorter than those generated by E. To our knowledge, the provers trained by our system are the only ones that use machine learning to match the performance of a state-of-the-art first-order logic prover *without already being based on one*. The experiments also demonstrate that HER allows the learner to improve more smoothly than when learning from proven conjectures alone. We believe that HER is a very desirable component of future ATP/ML systems.

Providing more side information to the transformer network, so as to make decisions more context-aware, should lead to further significant improvements. More importantly, while classical first-order logic is a large and important formalism in which many conjectures can be expressed, various other formal logic systems have been developed which are either more expressive or more adapted to different domains (temporal logic, higher-order logic, FOL with equality, etc.). It would be very interesting to try our approach on domains from these logic systems. While the given-clause algorithm and possibly the input representation for the neural networks would need to be adapted, the rest of our methodology is sufficiently general to be used with other logic systems, and still be able to deal with domains without known proofs.

Reproducibility Statement

The details of how the dataset is created and divided in different domains is explained in Section 3. Additionally, we include the list of problem files and axiom files used in each domain in the attached supplementary material. The variant of DISCOUNT algorithm is provided in appendix B. The heavy tail distribution used to sample the hindsight goals is provided in appendix C. The details of the hyperparameters used for all the experiments are included in appendix D. The training code was written in JAX (Bradbury et al., 2018). For training, we use 1 CPU for each actor and 1 NVIDIA V100 GPU for each learner.

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

## A  A QUICK PRIMER ON FIRST-ORDER LOGIC AND RESOLUTION CALCULUS

First-order logic (FOL) is a formal language used to express mathematical or logical statements. We give a brief introduction to first-order logic here. For more information, see (Fitting, 2012). Any statement expressed in first-order logic is called first-order logic formula. For example the statement: "for all people $X, Y$ and $Z$, if $X$ is a parent of $Y$ and $Y$ is a parent of $Z$, then $X$ is grandparent of $Z$" can be expressed as the FOL formula: $\forall X, \forall Y, \forall Z, \mathrm{parent}(X,Y) \wedge \mathrm{parent}(Y,Z) \Rightarrow \mathrm{grandparent}(X,Z)$. Here, $X, Y, Z$ are *variables*, and parent and grandparent are *predicates*. We will only consider FOL formulas expressed in Conjunctive Normal Form (CNF), as a conjunction ($\wedge$) of *clauses*, in which all variables are implicitly universally quantified ($\forall$). Consider the following CNF formula:

$$\underbrace{(\neg\mathrm{parent}(X,Y) \vee \neg\mathrm{parent}(Y,Z) \vee \mathrm{grandparent}(X,Z))}_{C_1} \wedge \underbrace{\mathrm{parent}(\mathrm{alice}, \mathrm{bob})}_{C_2} \wedge \underbrace{\mathrm{parent}(\mathrm{bob}, \mathrm{charlie})}_{C_3}$$

A *clause* is a disjunction ($\vee$) of a number of *literals*; we will also consider clauses to be sets of literals to simplify the notation. $C_1, C_2$ and $C_3$ are clauses. Note that $C_1$ is equivalent to the first FOL formula example above, expressed as a clause. A *literal* is an *atom*, possibly preceded by the negation $\neg$, in which case it is a negative literal (positive otherwise). For example, $\mathrm{parent}(X,Y)$ and $\neg\mathrm{parent}(Y,Z)$ are literals. An *atom* is a *predicate* name of arity $n \in \mathbb{N}$ followed by a list of $n$ *terms*. A *term* is either a *function* name of associated *arity* $n \in \mathbb{N}$ followed by a list of $n$ terms, or a *constant* (such as alice, bob), or a *variable*. We assume that two different clauses of the same formula cannot share variables. A clause $C$ is a *tautology* if $C$ contains both a literal $\ell$ and its negation $\neg\ell$. Tautologies can be safely removed from a CNF formula without changing its truth value. The `tree_size` of a clause is the number is the number of nodes when the clause is represented as a tree of terms. For example, `tree_size`$(\mathrm{parent}(X, \mathrm{bob}))$ is 3.

A *substitution* is a set $\{V_1 \rightsquigarrow t_1, V_2 \rightsquigarrow t_2, \dots\}$ where $V_1, V_2, \dots$ are variables and $t_1, t_2, \dots$ are terms. The set of variables $\{V_1, V_2, \dots\}$ is the *domain* of the substitution. The *application* $\sigma(C)$ of a substitution $\sigma$ to a clause $C$ (or also to a single literal) results in a new clause $C' = \sigma(C)$ where all occurrences of the variables (of the domain of $\sigma$) in $C$ are replaced with their corresponding terms according to the substitution $\sigma$. For example, if $C = \mathrm{parent}(X,Y)$ and $\sigma = \{X \rightsquigarrow \mathrm{alice}, Y \rightsquigarrow Z\}$ then $\sigma(C) = \mathrm{parent}(\mathrm{alice}, Z)$.

Two literals $\ell_1$ and $\ell_2$ can be *unified* if there exists a substitution $\sigma$ such that applying it to both literals result in the same literal, that is, $\sigma(\ell_1) = \sigma(\ell_2)$. In such a case, the most general unifier $\mathrm{mgu}(\ell_1, \ell_2)$ of $\ell_1$ and $\ell_2$ is the smallest substitution that unifies the two literals, and it is unique (up to a renaming of the variables). For example, the most general unifier of $C = \mathrm{parent}(X,Y)$ and $C' = \mathrm{parent}(\mathrm{alice}, Z)$ is $\{X \rightsquigarrow \mathrm{alice}, Y \rightsquigarrow Z\}$, such that $\sigma(C) = \sigma(C') = \mathrm{parent}(\mathrm{alice}, Z)$.

A clause $C_1$ *subsumes* a clause $C_2$ if there exists a substitution $\sigma$ such that $\sigma(C_1) \subseteq C_2$ where the clauses are considered to be sets of literals.[6] For example the clause $p(X, a)$ subsumes the clause $p(b, a) \vee p(c, a)$, with $\sigma = \{X \rightsquigarrow b\}$ (or also with $\sigma = \{X \rightsquigarrow c\}$) as $\sigma(\{p(X, a)\}) \subseteq \{p(b, a), p(c, a)\}$.

The clause $C'$ is a *factor* of a clause $C$ if there exist a substitution $\sigma$ and two literals $\ell$ and $\ell'$ in $C$ such that $\sigma = \mathrm{mgu}(\ell, \ell')$ and $C' = \sigma(C \setminus \{\ell\})$. The operation $\mathrm{factoring}(C)$ returns the set of all factors (unique up to renaming of the variables) of $C$, with 'fresh' (never used) variables. For example, we can factor $C_1$ on its first two literals to obtain the clause $\neg\mathrm{parent}(Y', Y') \vee \mathrm{grandparent}(Y', Y')$ with the substitution $\{X \rightsquigarrow Y, Z \rightsquigarrow Y\}$. A clause is the sole *parent* of its factors.

The clause $C''$ is a *resolvent* of two clauses $C$ and $C'$ if there exist a substitution $\sigma$, a positive literal $\ell$ in $C$ and a negative literal $\ell'$ in $C'$ such that $\sigma = \mathrm{mgu}(\ell, \ell')$ and $C'' = \sigma(C \setminus \{\ell\} \cup C' \setminus \{\ell'\})$. The operation $\mathrm{resolution}(C, C')$ produces the set of all possible resolvents of $C$ and $C'$ (Robinson, 1965), with 'fresh' variables. For example, the resolvents of $C_1$ and $C_2$

---

[6] We assume that syntactic duplicate literals are removed automatically.

are $\{\neg\text{parent}(\text{bob}, Z') \vee \text{grandparent}(\text{alice}, Z'), \neg\text{parent}(X', \text{alice}) \vee \text{grandparent}(X', \text{bob})\}$. The clauses $C$ and $C'$ are called the *parents* of $C''$. The *ancestors* of a clause are its parents, the parents of its parents and so on.

Together, resolution and factoring are sound and also sufficient for *refutation completeness*, that is, they can only produce clauses that are logically implied by the initial clauses, and if the empty clause is logically implied by the initial clauses, then the empty clause can be also be produced by a sequence of resolution and factoring operations starting from the initial clauses. For example, suppose that we want to prove that alice is the grandparent of someone, that is, that $\text{grandparent}(\text{alice}, A)$ can be satisfied for some value of $A$. Then we negate this conjecture to obtain the clause (implicitly universally quantified over $A$) with a single literal: $C_4 = \neg\text{grandparent}(\text{alice}, A)$ and we attempt to refute the CNF formula $C_1 \wedge \cdots \wedge C_4$, that is, to reach the empty clause using resolution and factoring. First we can resolve $C_4$ with $C_1$ to obtain $C_5 = \neg\text{parent}(\text{alice}, Y') \vee \neg\text{parent}(Y', A')$. Then we can resolve $C_5$ with $C_2$ to obtain $C_6 = \neg\text{parent}(\text{bob}, A'')$ and finally we can resolve $C_6$ with $C_3$ to obtain the empty clause, which means that indeed alice is the grandparent of someone.

## B  OUR DISCOUNT-LIKE ALGORITHM

Our simple search algorithm is given in Algorithm 3. See the main text for more details.

**Algorithm 3** Our variant of the DISCOUNT algorithm. Note that we use three priority queues for the candidates (see main text).

```python
def search(input_clauses)
    candidates = input_clauses
    active_clauses = {}
    # Saturation loop.
    while candidates is not empty:
        given_clause = extract_best_clause(candidates)
        if given_clause is empty_clause: return "refuted"
        if tautology(given_clause): continue  # discard the given_clause
        for c in active_clauses:
            if c subsumes given_clause: continue # forward subsumption
            if given_clause subsumes c: active_clauses.remove(c) # backward
                subsumption
        # Generate factors and resolvents.
        candidates.append(given_clause.factors())
        for c in active_clauses:
            candidates.append(resolve(given_clause, c))
        active_clauses.insert(given_clause)
    return "saturated"
```

## C  A HEAVY-TAIL DISTRIBUTION OVER THE INTEGERS

To ensure a preference for smaller clauses, while ensuring some diversity of the clause sizes, we use the following heavy-tail distribution for $s \in \{0, 1, 2 \dots\}$:

$$w_s = 1/\ln(s + e) - 1/\ln(s + e + 1).$$

These weights constitute a telescoping series and ensure that $\sum_{s=0}^{\infty} w_s = 1$ while, using $\ln(1 + 1/x) \geq 1/(x + 1)$,

$$w_s = \frac{\ln\left(1 + \frac{1}{s+e}\right)}{\ln(s + e)\ln(s + e + 1)} \geq \frac{1}{(s + e + 1)(\ln(s + e + 1))^2}.$$

Thus, $w$ is a heavy-tailed universal distribution over the nonnegative integers in the sense that $-\ln w_s \in O(\ln s)$, similarly to Elias' delta coding (Elias, 1975).

Tangentially, sampling according to $w_s$ is simple since its cumulative distribution telescopes: Sample $u$ uniformly in $[0, 1]$, then select the integer $\min\{s \geq 0 : 1 - 1/\ln(s + e + 1) \geq u\}$, that is, select $s = \lceil \exp(1/(1 - u)) - e - 1 \rceil$.

## D  HYPERPARAMETERS

For the transformer encoder, hyperparameter notations from Vaswani et al. (2017) are given in parenthesis for reference. The model is trained using stochastic gradient descent with the Adam optimizer (Kingma & Ba, 2015) with $\beta_1 = 0.9$, $\beta_2 = 0.999$ and $\epsilon = 10^{-8}$. A subset of hyper-parameters have been selected by running the model on the FLD1 domain, selected values are underlined: number of layers ($N$) in $\{\underline{3}, 4, 5\}$, embedding size ($d_k$, $d_v$) in $\{\underline{64}, 128, 256\}$, hash vector size in $\{16, \underline{64}, 256\}$, learning rate in $\{0.003, \underline{0.001}, 0.0003\}$, probability of dropout ($P_{drop}$) in $\{0., \underline{0.1}, 0.2, 0.3, 0.5, 0.7\}$. The other hyperparameters were fixed: the batch size is 2560, the number of attention heads ($h$) is 8, which leads to a dimensionality ($d_{model}$) of 512, the inner-layers have a dimensionality ($d_{FF}$) of 1024. To ensure diversity in the training examples, learners wait for the experience replay buffer to contain at least 65536 examples and then sample uniformly from it. The learners used Nvidia V100s GPUs with 16GB of memory.

For the actors, the age, weight, and learned-cost queues in our given-clause algorithm are selected on average 1/13th, 3/13th and 9/13th of the steps, respectively. The batch size is set at 320. All actors and E are limited at 8GB of memory on modern AMD 64bit platforms.

