# OpenReview forum: "Proving Theorems using Incremental Learning and Hindsight Experience Replay"
_ICLR.cc/2022/Conference — ICLR 2022 Submitted_

### Official Review · Reviewer_ywZc · 2021-10-28

**Correctness:** 3
**Technical Novelty And Significance:** 3
**Empirical Novelty And Significance:** 3
**Recommendation:** 8
**Confidence:** 4

**Main Review:**

Strengths:
- The paper tackles an important problem in computer science: automated theorem proving, and more specifically seeks to resolve the data-scarcity issue that prevents large neural networks from being applied to this area
- the proposed approach extends hindsight experience replay to the domain of theorem proving. While not controlled in this experiment, the use of length-based bucketing and sampling ensure that the overwhelming amount of training data produced by the relabelling do not overwhelm the system, nor over-emphasize long (later) goal proofs.
- The proposed approach presents strong results that compete with E, and improve greatly over the two baselines: original system, and original system w/o HER.
- the spectral representation is novel. Its utility is harder to asses without ablations, but given a code release, building upon this work and performing further experiments could prove fruitful as well
- the use of a tabula rasa approach (purely incremental) is remarkable, and bodes well for adaptability to new domains where we lack starting theorems or data


Weaknesses
- For learnt automated theorem proving systems to become practically useful they would need to outperform or be complementary to existing systems. The proposed approach, while providing shorter proofs, fails to outperform E in terms of number of theorems proven. It may be valuable to include an experiment where the results from E can be combined with those from the proposed system to see if a union is actually better
- Experiments, while costly, are not repeated with confidence intervals and uncertainties -- thus, while the results are compelling, it is hard to rule out whether the specific ordering or randomness of the search process could impact the results in a strong way
- certain design decisions such as size of goal sampling are not ablated, and while the comparison between HER and not HER is excellent in Figure 2, such as comparison at a smaller scale would be welcome.
- Similarly the importance of "a separate network is trained per domain" is not measured, and this decision could be impactful when designing future experiments.


**Summary Of The Paper:**

The authors present an incremental learning (self-bootstrapped), Hindsight-Experience-Replay-based ATP learner, that learns to reordering candidates in a queue during a search process. The key insight is the data scarcity of ATP, wherein most searches do not lead to the goal, while the use of HER enables the use of failed searches as positive examples of alternate goal proofs

**Summary Of The Review:**

The authors propose a very compelling solution to the data scarcity problem in automated theorem proving: relabelling failed searches by the sub-goals reached. Several important details for getting this to work are laid out, and the system is able to outperform baselines and compete with the existing E prover. The main shortcoming of the paper is a lack of repeat experiments, and missing desirable ablations for high-level design decisions.

---

### Official Review · Reviewer_MLCG · 2021-11-01

**Correctness:** 3
**Technical Novelty And Significance:** 2
**Empirical Novelty And Significance:** 3
**Recommendation:** 5
**Confidence:** 5

**Main Review:**

Strength:
=======
- It is clear that this work took a substantial engineering effort to produce the experimental results.
- The results are encouraging at least on the dataset that the authors tested their thesis on, which motivates further exploration on applying HER in ATP domain.
- The proposed use of spectral encoding in this context can be a useful contribution however no experimental results were reported to ablate that [see below].

Weakness:
=========
- The idea of using HER, although novel in the context of theorem proves, is a fairly immediate idea to apply in any search setting with sparse signal.
- The use of spectral encoding is not properly justified and no ablation is performed to compare its superiority to more conventional means of formula representation, e.g., GNNs. Of course the choice of using spectral encoding over GNNs can be regarded as merely a design decision that seems to work in this context but since it was explicitly positioned as a contribution, in both introduction as well as the conclusion section, it merits more empirical elaboration.
- Head-to-head comparison of IL against E in terms of wall-clock time could be misleading in that E is running on a single core per conjecture whereas IL is benefitting from 1000 actors running in parallel (presumably on 1k cores). This criticism does not lessen the validity of the contribution, as it can be argued that IL enables parallelism in provers, however including a comparison in terms of FLOPS or total CPU time (even in the appendix) can paint a clearer picture.
- The premise of the manuscript is that reliance of most ML-based methods on high-end provers limits their potential of surpassing human capabilities. The proposed technique might be a solution for that issue particularly in cases where generating training data is either expensive or impossible. In the particular case of TPTP however, we **are** capable of generating the training set. Indeed E manages to prove almost all of the instances at around the $10^4$ second mark. So why not train a model (with the same architecture) on this training set and compare it against IL+HER as a baseline? That would further strengthen the authors’ claims.

Questions:
========
- What is the justification for using 10 learners? Is it to spread the load of processing new samples across them or is it to introduce some diversity in the clause scoring?

Typos:
=====
- [Page 2 - subsection (2.1)] - 1st sentence: The word “simple” is repeated twice: “... we opted for a simple simple baseline...”

**Summary Of The Paper:**

This paper proposes an “incremental learning” algorithm for training a distribution (domain)-specific theorem prover for FO formulae without the need of a classical prover to generate training examples (intermediate clauses of a proof). Given a set of axioms and the negation of the (to be proven) conjecture, the prover iteratively 1. finds factors of clauses; and 2. applies resolution, until either an empty clause is reached (conjecture is proven) or no more new clauses can be generated (the conjecture is dis-proven). It is important for the prover to select clauses for factorization and/or resolution, that would lead to an empty clause faster. To that end, the AI-based theorem provers are augmented with a ML model (policy) that assists the prover by scoring the candidate clauses based on the probability of their applicability in building a proof. This in turn leads to construction of shorter proofs. The authors incorporate Hindsight Experience Replay (HER) in order to mitigate the lack of expert (machine and/or human generated) training sets. They show that HER indeed helps their incremental learning scheme to a point that the resulting ML-aided prover achieves comparable performance wrt. to the SOTA conventional prover E. Additionally the authors propose the use of spectral encoding to represent formulae instead of positional encoding of transformers.

The resulting architecture is tested on TPTP benchmark against E, along with an ablation to show the effectiveness of HER. The comparison with E is to measure both the number of proven conjectures within a cutoff as well as the proof quality (proof length).


**Summary Of The Review:**

The paper demonstrates an experimental signal in support of using HER for bootstrapping ML-based ATPs without expert labeled data on TPTP dataset. The reliance of ML-based provers on expert data is a serious issue and efforts to alleviate that reliance are definitely encouraged. However just showing the effectiveness of a well-known technique such as HER, which perhaps is one of the most immediate remedies that comes to mind for sparse reward tasks, seems like an incremental contribution, especially since some critical baselines are missing which could’ve made the claims stronger. Note that incremental learning has been used in other shapes or forms before in this context and thus the main value proposition of the manuscript is the addition of HER to that scheme.

---

### Official Review · Reviewer_iBNz · 2021-11-02

**Correctness:** 3
**Technical Novelty And Significance:** 2
**Empirical Novelty And Significance:** 3
**Recommendation:** 5
**Confidence:** 3

**Main Review:**

Including the primer on first order logic is an excellent idea, thank you.

Overall the writing is clear and well structured. The presentation of the figures and tables is a pleasure to behold.

The title of the paper seems somewhat misleading given that this is not RL but an HER inspired incremental learning method.

The notation in appendix C is unusual for an ML audience. I would recommend to write something like p(X=k)=... instead of stating w_s which we must infer is the value of the pmf (e.g. copy the notation of https://en.wikipedia.org/wiki/Poisson_distribution).

TacticZero by Wu et al 2021 is missing in the references. That work includes an RL reward for sub goals which are proved, so is highly related to hindsight experience replay in the sense that attempts that do not prove a main goal nonetheless serve as training data. Moreover by training end to end the scoring functions of Wu et al are properly coupled to the search process. This is not the case for the submission, where the training of the scoring is not optimally matched with the search process.

The experimental performance is reasonable. Losing out to E suggests that the heuristics of E could be included as features for the scoring rule - though it is not essential to add that here of course, this is just a suggestion.

**Summary Of The Paper:**

The paper applies hindsight experience replay (HER) to automatic theorem proving (ATP). ATP attempts to prove logic statements by (roughly speaking) showing that there are no counterexamples, i.e. showing that nothing resolves the negation of a statement. HER is a technique from reinforcement learning which mitigates sparse rewards by treating failed attempts as successful attempts on a different problem, namely that with the end state of the attempt as the goal.

The paper is not quite an HER method however ; it does not use reinforcement learning but rather incremental learning, where clauses generated during proof attempts are used as additional data for (incremental) supervised learning.

**Summary Of The Review:**

This is a nicely presented work with a novel approach to using clauses generated during proof search for training. The approach is somewhat ad hoc however, and could benefit from a principled RL scheme which learns an optimal policy, rather than fixing a heuristic search and combining that in an ad hoc way with a scoring rule.

---

### Official Review · Reviewer_8LVa · 2021-11-02

**Correctness:** 3
**Technical Novelty And Significance:** 2
**Empirical Novelty And Significance:** 3
**Recommendation:** 6
**Confidence:** 4

**Main Review:**

Strength: the idea of using hindsight experience replay is straightforward (which is good). The algorithms are well-thought-out (such as using time scheduling and subsampling the hindsight goals) and the experiments clearly show the gain of using hindsight goals.

Weakness: my major concern is the absolute performance of the resulting prover. Table 1 suggests that the resulting prover is marginally better than E — being able to prove 1 or 2 more theorems in some domains, while E finds ~10 more theorems on FLD1 and FLD2. It is fine that the resulting prover does not outperform E which is SOTA, but in that case I would like to see at least one comparison with other learning based approach, evaluated on either TPTP or another suitable data set.

Table 1 also seems to suggest that TPTP is probably too easy for both E and IL w/HER. Have you considered using other benchmarks (e.g., [GRUNGE](https://arxiv.org/abs/1903.02539))?

Also note that [TacticZero](https://arxiv.org/abs/2102.09756) uses policy gradient which implicitly uses subgoals in failure proof attempts to train the neural networks.

Minor:
1. The representation seems a bit too complicated. Would a graph neural network make life easier? Or perhaps simply a transformer encoder.
2. “Although the set of active clauses is an important factor … we ignore it …”. This is probably the right place where you can introduce hand-engineered features that summarize the information of the set of active clauses.
3. Since the hindsight goals probably contain the information similar to the original goal, I suspect that HER wouldn’t offer too much when evaluated on goals from another domain. I would like to see a bit more cross-domain evaluation, but since the paper already made it clear that the approach is domain-specific, the current experiments are fair.

**Summary Of The Paper:**

The paper proposes an incremental learning algorithm for training domain-specific provers for first-order logic without equality, and addresses the sparsity of training data by adapting hindsight experience replay. In short, it uses clauses generated during a proof attempt to generate new goals that can be explored further.

**Summary Of The Review:**

Although the absolute performance of the proposed framework is not at a state-of-the-art level, the usage of HER (though not entirely novel) is justified and the overall algorithm is novel and well-designed. It is unclear how the framework compares to other learning-based systems. The paper is well-written.

---

### Official Review · Reviewer_89SU · 2021-11-04

**Correctness:** 3
**Technical Novelty And Significance:** 2
**Empirical Novelty And Significance:** 2
**Recommendation:** 3
**Confidence:** 4

**Main Review:**

Strengths:

The idea of creating positive examples from failed proofs seems new in theorem proving and, together with HER, shown to significantly improve performance of the learning-based prover.

They propose a transformer based representation of logical statements.

Weaknesses:

It's hard to compare the proposed approach to other learning-based approaches in the literature. The authors create their own dataset instead of using benchmarks widely used in the literature (M2k [1] and MPTP2078[2]). This is probably due to their underlying prover not supporting entire first-order language as it cannot solve problems with the equality symbol. This can be addressed by replacing their underlying prover with any SOTA classical prover (e.g., E prover) but disabling its heuristics as done in some work in the literature (e.g., [3] uses Beagle as underlying prover with its heuristics disabled).

The paper seems to downplay the performance of the approach proposed in [3] by saying it still doesn't get close to E. But the only reason they can say theirs is competitive against E is because their dataset is classified into domains, where we can clearly see their approach beating E in some domains but on aggregate results their approach still falls short to beat E, just like [3]. Furthermore, their subset of TPTP problems without equality and number of axioms less than 1000 could be much easier than problems with equality and have significantly more axioms.

Provers in their experiments are not given a time limit for attempting to prove theorems as typically done in the literature (100 seconds is used in [1] and [3]). Again, this makes it possible to evaluate the proposed approach in the context of related work.

They proposed a new representation of logical statements based on transformers but a motivation of this new representation is not given, especially given that graph neural networks have been shown to represent logical statements well. Also, the benefit of such a representation is not shown empirically. So we don't know if this proposal adds any new value.

It's not clear if the incremental learning approach used should, in principle, be better than reinforcement learning based approaches used in the literature (e.g., [3]). In principle both approaches are learning from scratch, but their approach learns a scoring network from examples generated as successful proofs (included those added from unsuccessful ones) whereas RL based approaches use a reward mechanism.

By the way, recent work on building machine learning based proof guidance systems have come closer to beating E on M2k dataset and actually beat it on MPTP2078 dataset [4].

[1] Kaliszyk, C.; Urban, J.; Michalewski, H.; and Olsák, M. 2018. Reinforcement learning of theorem proving. In Advances in Neural Information Processing Systems 31, NeurIPS 2018, 8836–8847.
[2] Alama, J.; Heskes, T.; Kühlwein, D.; Tsivtsivadze, E.; and Urban, J. 2014a. Premise selection for mathematics by corpus analysis and
kernel methods. Journal of Automated Reasoning 52(2):191–213.
[3] Maxwell Crouse, Ibrahim Abdelaziz, Bassem Makni, Spencer Whitehead, Cristina Cornelio, Pavan Kapanipathi, Kavitha Srinivas, Veronika Thost, Michael Witbrock, and Achille. Fokoue. A deep reinforcement learning approach to first-order logic theorem proving.
Proceedings of the AAAI Conference on Artificial Intelligence, 35(7):6279–6287, 2021
[4] Ibrahim Abdelaziz, Maxwell Crouse, Bassem Makni, Vernon Austil, Cristina Cornelio, Shajith Ikbal, Pavan Kapanipathi, Ndivhuwo Makondo, Kavitha Srinivas, Michael Witbrock, Achille Fokoue, https://arxiv.org/abs/2106.03906#

**Summary Of The Paper:**

The paper proposes an incremental learning approach to theorem proving that learns to prove theorems from scratch, using Hindsight Experience Replay (HER) to learn from unsuccessful proofs. The approach adopts ideas from goal-conditioned reinforcement learning and creates more training examples by turning intermediate steps of an unsuccessful proof attempt into proved conjectures and add these into positive examples to learn from. This significantly increases the number of examples to learn from compared to related work that only learns from successful proofs. The approach is shown to be competitive against E prover, a SOTA prover, on a subset of TPTP without equality.

**Summary Of The Review:**

While the idea of using unsuccessful proof attempts with HER seems new and can benefit the learning-based theorem proving community, as it stands, the paper doesn't do a good job of showing the benefits of their overall theorem prover empirically. Combining these ideas with traditional ATPs and evaluating on published benchmarks is recommended.

---

### Author Response · Authors · 2021-11-17
**Authors' Response to the Reviewers (Part 1)**

First of all, we would like to thank all the reviewers for their detailed comments and helpful feedback. This is very appreciated. Note that, due to the character limits, we split our response into multiple parts. We also group our
responses by concerns to prevent repetition.

### Comparison to other learning-based approaches

Reviewers 89SU and 8LVa wondered about a comparison with other learning-based approaches of the literature.

We believe that we partly address this need by providing a learning-based baseline (IL w/o HER): The method that we use in this baseline is a very popular one and represents a significant part of the literature (see, for example, Loos 2017
and Chvalovský 2019). It is true that we do not compare our method against specific RL-based approaches. However, we would like to stress that the main contribution of this work is about assessing whether using our take on hindsight
experience replay (HER) makes a difference when comparing with the same base prover. Using a simplistic resolution prover ensures that there is less interference with all the various tricks that are used in optimized provers and our
findings are that HER does improve significantly over learning only from proofs found on actual problems. Our hope is that other research teams will consider also using HER with the (smarter) base prover, whether using RL or not. As such, a
comparison with other provers, while interesting, is somewhat orthogonal to the question tackled in this paper.

Moreover, to perform a fair comparison within our setup with other ML provers would require reimplementing them, which is a significant technical endeavor (e.g., the code for the most interesting algorithm, TRAIL, is not available).

### Representation: Why transformers and not graph neural networks?

Reviewer 89SU, MLCG and ywZc raise the fair point that the choice of the formula representation and the transformer-based architecture associated with it are not properly justified.

We indeed considered other representations and architectures, specifically a multi-layer perceptron with handcrafted features and transformers with a sequential representation with positional encoding. The overall results are given below.

| Conjectures | MLP | Transformer (Sequential) | Transformer (Spectral) |
|-------------|-----|--------------------------|------------------------|
| 1116        | 778 | 869                      | 949                    |

We are happy to add these results to the appendix. This partly addresses the concern regarding the justification of the use of the spectral features (especially in comparison with the sequential representation).

Specifically for GNNs, we ran some preliminary experiments in which GNNs achieved slightly worse results. The reason we did not include a comprehensive ablation study was due to technical difficulties around running full-scale
experiments with GNNs.

Given the general agreement among the reviewers, we have removed the spectral approach from the list of contributions as stated in the introduction (paragraph starting with "In this paper").

### Adapting Hindsight Experience Replay to Automated Theorem Proving

Reviewer 8LVa and MLCG stated that using hindsight experience replay is straightforward.

Although we agree with the reviewers that the idea of using hindsight experience replay (HER) is somewhat obvious in domains where reward is sparse, we believe that the adaptation of the idea specifically to theorem proving was not so
trivial. In our paper, we show how clauses reached can be turned into goals in a principled way in the case of first-order formulas. We also show a theoretically sound way to deal with the abundance of goals in theorem proving.

Furthermore, in spite of the significant improvements we observed when using hindsight, we have not seen any mention of using it in the theorem proving literature.

### Our method vs. reinforcement learning

Reviewer iBNz suggested using a more principled reinforcement learning (RL) scheme.

We believe that the question we answer in the paper is orthogonal to the question of how to apply a principled RL approach to automated theorem proving (ATP). We are instead focused on two other aspects of the ATP challenge: (i) how to
make use of failed attempts, and (ii) how to allocate resources when given a pool of conjectures to prove (applying uniform budgeted scheduler to this domain). We provide solutions to these problems and demonstrate that they work even in
the absence of hand-crafted heuristics. The solutions we provide can easily be reused in an RL-based approach.

We would also like to emphasize that our approach to learning is far from being improvised; it is indeed based on other important work in literature (e.g. Loos 2017 and Chvalovský 2019).

---

> ### Author Response · Authors · 2021-11-17
> **Authors' Response to the Reviewers (Part 2)**
>
> ### Dataset: Why TPTP and not M2k, MPTP2078 or GRUNGE?
>
> Reviewer 89SU and 8LVa both pointed out that there are other benchmark datasets that we could have used (Mizar, GRUNGE).
>
> As appreciated by the reviewers, we are unable to use the mentioned datasets (nor the rest of TPTP) due to the limitations of our base prover (mainly the lack of support for equality). The reason we decided to use a simple base prover and
> not depend on an advanced prover like E was to test the ability of our approach to train a skeleton prover without taking advantage of _any_ of the optimizations present in mature provers. Many other learning methods that were integrated
> into provers like E end up taking advantage of heuristics unrelated to clause scoring, such as premise selection, literal selection, advanced redundancy elimination, special data structures, etc. A number of them cannot be disabled to
> obtain a prover that is as simple as ours.
>
> That said, we of course do believe in the value of integrating our methods with more powerful base provers, in particular to obtain results on a wider variety of benchmarks, but this is left as future work.
>
> ### Time limit cap
>
> Reviewer 89SU said in their review that we are not using a time limit for our prover's attempts.
>
> We actually do use a maximum time limit of one hour for each attempt made by our prover. It is our mistake to not emphasize this properly in the paper. In the updated version, we emphasize this time limit of one hour in both the
> methodology ("Time scheduling" paragraph) and the experiments sections (paragraph starting with "To compare our results"). We thank the reviewer for pointing this out.
>
> ### Comparison to TRAIL
>
> Reviewer 89SU suggested that we are downplaying the performance of TRAIL (Crouse 2021) in our treatment of previous work. They also refer to the more impressive results in the newest iteration of TRAIL (Abdelaziz 2021).
>
> Indeed, we do state in our paper that TRAIL (as described in Crouse 2021) fails to reach state-of-the-art performance, and the same is true for our method in the overall results (though see below). Note, however, that the evaluation setup
> used in both of the TRAIL papers (Crouse 2021 and Abdelaziz 2021) is different than ours: E is given a single attempt with a time limit of 100 seconds whereas TRAIL is run for many times on the dataset (in incremental learning or standard
> evaluation setting, which have their best results). In the end, only one run with a time limit of 100 seconds is used to compare TRAIL to E. By contrast, in our setup, we consider the whole training time as a part of the search time and
> compare accordingly: We run E on each problem for the same amount of wall-clock time (i.e. seven days) as we use to train our prover.
>
> To provide results that would help compare to the literature, we updated our paper to include results from E with a one hour time limit (i.e. the maximum time limit we use for each attempt of our prover) which is a commonly used time limit.
> As can be seen in the updated Table 1, our prover proves 14 more problems than E, when E is run with a time-limit of one hour, same as in our prover. By this measure, our method outperforms E.
>
> | Conjectures | IL w/HER (max. one hour per attempt) | E (one hour per attempt) | E (seven days per attempt) |
> |-------------|--------------------------------------|--------------------------|----------------------------|
> | 1116        | 949                                  | 935                      | 972                        |
>
> ### Relations to TacticZero
>
> We thank Reviewer 8LVa and iBNz for the TacticZero reference (a concurrent work in higher-order logic), which, as we understand, is going to be published at NeurIPS 2021. We will make sure to cite it in the camera-ready version of our
> paper.
>
> ### Other concerns
>
> **Title of the paper**
>
> We did not anticipate that this would be misleading, as we do not mention reinforcement learning (RL) in the title but only hindsight experience replay (which can be applied independently of RL). Furthermore, our methodology can be
> used in an RL-based learner. Nevertheless, we are happy to consider alternative titles if the reviewer and other reviewers feel strongly about this issue.
>
> **Comparison of IL against E in terms of wall-clock time could be misleading**
>
> As stated by the reviewer, E is inherently not parallelizable and our distributed setup can scale to 1000 actors, so we included the comparison with wall clock time. However, we have clarified this point explicitly in the
> manuscript to avoid any confusion ("Training vs. searching" paragraph in the experiments section).
>
> **What is the justification for using 10 learners**
>
> Using multiple learners learning from a single source of experience is an inexpensive way of adding some diversity without increasing the acting budget. We have clarified this point in the paper ("Distributed implementation" paragraph
> in the methods section).

---

> > ### Author Response · Authors · 2021-11-17
> > **Authors' Response to the Reviewers (Part 3)**
> >
> > ### Other concerns (continued)
> >
> > **Experiments, while costly, are not repeated with confidence intervals**
> >
> > It is true that the experiments are not shown with confidence intervals as most of the experiments are computationally expensive. We have verified the results do not significantly differ in at least a couple of runs. However, we will
> > include the confidence intervals with five runs in an eventual final submission. Furthermore, most of the stochasticity in learning comes from the random initialization of the networks. As we are using ten learners and as we are measuring
> > the total number of conjectures proved by any of the ten models, the variance we observe is extremely small.
> >
> > **Certain design decisions such as size of goal sampling are not ablated**
> >
> > Running experiments is rather expensive, hence we focused on the most important ablation studies. We did not spend too much time tuning these parameters and used common defaults or our best initial guess.
> >
> > **Using hand-engineered features, heuristics of E and combining results of E and our approach**
> >
> > We agree with the reviewer that the heuristics of E could be added as additional features for scoring the clauses and one can also use hand engineered features for the active set. However, in this work, we illustrate that a pure
> > learning system augmented with hindsight experience replay can achieve results significantly closer to a well-tuned heuristic system with no additional support, unlike previous works. However, we can point out that there are nine
> > theorems which are proven by our system but not by E. Hence, we definitely expect the results to improve further when this system is integrated with E. The results by taking a simple union of theorems proven by IL w/HER and E are given
> > below.
> >
> > | Conjectures | IL w/HER | E   | Union(E, IL w/HER) |
> > |-------------|----------|-----|--------------------|
> > | 1116        | 949      | 972 | 981                |
> >
> > Thanks again for the reviews.

---

> ### Comment · Reviewer_89SU · 2021-11-29
> **Response to author responses**
>
> Thank you for the authors' responses.
>
> It's true that the experiments in the paper show the benefit of HER when used with a simple incremental learner. But how much of this is due to the fact that the learner is too simple to solve most problems? And what guarantees that this would easily generalise to a learner that already performs better than this simpler prover? The only way to truly know is to use HER across multiple provers, from your simple one to others in the literature. Without this, the paper looks like a good idea that has not been shown to work in general. The argument that HER hasn't been applied to ATP before is weak given that the paper doesn't show it works in general. For all we know, it has been tried but was not published because the results were not significant enough.
>
> I really get the idea of using a simple prover to avoid interference from heuristics in SOTA provers. However, I think showing the benefit of HER on published benchmarks that require these SOTA provers and using a simple prover for ablation would greatly increase the strength of the paper.
>
> It's good to see that their proposed system outperforms E when given time limit of one hour. But bear in mind that this is only for problems with less than 1000 axioms, which are relatively easy. One of the main challenges of ATPs is identifying a small set of relevant axioms from a large set for a specific conjecture. This is where heuristics of E are expected to be advantageous.

---

> > ### Author Response · Authors · 2021-12-08
> > **Response to Reviewer 89SU**
> >
> > Thank you for your response.
> >
> > We agree that premise selection is an important component of building a theorem prover and that our approach does not solve that problem.
> >
> > We address the problem of using selected premises to arrive at the proof, which is an orthogonal but essential component too.
> >
> > Note that many of the previously published benchmarks like MPTP2078 (used in other papers such as TRAIL) have also used far fewer axioms than 1000 axioms used in this work. The bushy version of MPTP2078, which is used in most of the previously published works, has a maximum of 532 axioms with an average of 47 axioms per problem. The major reason for evaluating on these problems is that even state-of-the-art provers (like E or Vampire) are not close to solving all the theorems even in this "limited" setting.  Solving problems that have a large number of  axioms (like chainy version of MPTP2078) is important and we believe that the solution will require both premise selection (which we do not address) and efficient search (which we do).
> >
> > Hence, we would also like to clarify that even when based on a simple base prover, using HER not only significantly improves performance but even manages to reach state of the art heuristic provers for a non-trivial set of problems, of course noting the limitations of the dataset (no equality). Therefore we believe that this is not a toy domain and matching the performance of state-of-the-art prover by non-trivial adaptation of HER in theorem proving is a significant contribution in itself.

---

### Decision · Program_Chairs · 2022-01-20

**Decision:**

Reject

**Comment:**

Four reviewers acknowledged the author's response and did not change their largely negative scores.  The one enthusiastic reviewer did not respond to the more negative reviewers and has not worked in the theorem proving area. The main problem with the paper seems to be that the reviewers were not convinced by the empirical results.  They felt that results should have been presented on more widely used benchmark datasets.